# How Can Malnutrition Affect Autophagy in Chronic Heart Failure? Focus and Perspectives

**DOI:** 10.3390/ijms22073332

**Published:** 2021-03-24

**Authors:** Giovanni Corsetti, Evasio Pasini, Claudia Romano, Carol Chen-Scarabelli, Tiziano M. Scarabelli, Vincenzo Flati, Louis Saravolatz, Francesco S. Dioguardi

**Affiliations:** 1Department of Clinical and Experimental Sciences, Division of Human Anatomy and Physiopathology, University of Brescia, 25123 Brescia, Italy; cla300482@gmail.com; 2Cardiac Rehabilitation Division, Scientific Clinical Institutes Maugeri (IRCCS), Lumezzane, 25065 Brescia, Italy; evpasini@gmail.com; 3Hunter Holmes McGuire Veterans Affairs Medical Centre (VAMC), Division of Cardiology, Richmond, VA 23249, USA; chenscarabelli@hotmail.com; 4Centre for Heart and Vessel Preclinical Studies at St. John Hospital and Medical Centre, Wayne State University, Detroit, MI 48202, USA; tscarabelli@hotmail.com; 5Department of Biotechnological and Applied Clinical Sciences, University of L’Aquila, 67100 L’Aquila, Italy; vincenzo.flati@univaq.it; 6Department of Medicine at St. John Hospital, Wayne State University, Detroit, MI 48202, USA; louis.saravolatz@ascension.org; 7Department of Internal Medicine, University of Cagliari, 09124 Cagliari, Italy; fsdioguardi@gmail.com

**Keywords:** malnutrition, autophagy, heart failure, chronic diseases, amino acids

## Abstract

Chronic heart failure (CHF) is a disease with important clinical and socio-economic ramifications. Malnutrition and severe alteration of the protein components of the body (protein disarrangements), common conditions in CHF patients, are independent correlates of heart dysfunction, disease progression, and mortality. Autophagy, a prominent occurrence in the heart of patients with advanced CHF, is a self-digestive process that prolongs myocardial cell lifespan by the removal of cytosolic components, such as aging organelles and proteins, and recycles the constituent elements for new protein synthesis. However, in specific conditions, excessive activation of autophagy can lead to the destruction of molecules and organelles essential to cell survival, ultimately leading to organ failure and patient death. In this review, we aim to describe the experimental and clinical evidence supporting a pathophysiological role of nutrition and autophagy in the progression of CHF. The understanding of the mechanisms underlying the interplay between nutrition and autophagy may have important clinical implications by providing molecular targets for innovative therapeutic strategies in CHF patients.

## 1. Chronic Diseases and Malnutrition: The Mortal Embrace

Chronic diseases (CDs) are characterized by chronic inflammation with accompanying increased catabolism and reduced anabolism. This imbalance is also known as “hypercatabolic syndrome” (HCS). Although the pathophysiology of HCS is multi-factorial and nonhomogeneous, all CD patients display altered ratios between catabolic (e.g., TNF-α, cortisol, catecholamine, glucagon, cytokines) and anabolic factors (e.g., insulin, insulin-like growth factors, and growth hormone). HCS severely impacts the body’s metabolism, causing a disproportion between nutritional supply and energy demands as protein breakdown exceeds synthesis, leading to muscle wasting and cellular energy impairment with resultant alteration in protein turnover [1,2,3,4]. In addition, HCS promotes insulin resistance which, in turn, reduces protein synthesis and impairs metabolism, thus reinforcing the protein-amino acid disarrangement [3,5]. In patients, these metabolic conditions are related to worse clinical prognosis, increased mortality (independent of primary pathology), and increased health-related expenditure [6,7].

Among the CDs, chronic heart failure (CHF) is a pathology with a substantial impact on public health and public economic consumption. The factors leading to CHF can be grouped into four categories: (1) common risk factors such as hypertension, ischemic injury, metabolic syndrome; (2) genetic heart disease; (3) mechanical changes, such as valve dysfunction; and (4) immune-related causes, including infections (bacterial and viral) as well as autoimmune reactions [8]. Like in other CDs, CHF patients are characterized by high levels of inflammation [9], HCS with a decrease in body weight, sarcopenia (defined as loss of skeletal muscle and physical performance), and muscle wasting [10]. Clinically, these patients are often significantly malnourished as a nutritional intervention is frequently lacking or overlooked.

According to the World Health Organization, malnutrition is a condition defined as a supply/demand imbalance at the cellular level between the nutrients/energy intake and the organism’s real demand for nutrients essential to ensure its correct development and maintenance [11]. This inadequate caloric and nutrient intake can lead to a decrease in muscle mass and immune system function with consequent deterioration in the quality of life, as demonstrated by a correlation between malnutrition and survival [12,13]. Malnutrition should be suspected by evaluation of the concentration of serum albumin [2,12,14], with hypoalbuminemia (albumin values < 3 mg/dL) considered to be the main marker of a poor nutritional status and an important prognosticator of increased morbidity and mortality in strokes [15] or after cardiovascular surgery [16].

In patients admitted for heart failure exacerbation, protein-energy malnutrition was associated with 2.5 times greater mortality, a 3-fold increase in cardiogenic shock, 1.5 times greater risk of acute kidney failure and respiratory failure, along with resultant increase in hospital length of stays and increased hospitalization costs [17].

Serum albumin deficiency is observed in about 30% of CHF patients [18], with or without concurrent anemia [10]. In addition to its association with higher in-hospital mortality rates in patients hospitalized with acute heart failure [19], including acute non-ischemic heart failure [20], hypoalbuminemia is an independent predictor of long-term mortality in CHF patients [19,21].

The impact of hypoalbuminemia on survival is not limited to heart failure with reduced ejection fraction as albumin levels were predictive of mortality in patients with heart failure with preserved ejection fraction [18]. Nonetheless, independent of admission albumin levels, in patients who improved albumin levels during hospitalization for acute decompensated heart failure, a lower risk of 1-year mortality and hospitalization was reported [22].

In CHF patients with severe muscle wasting, adequate protein-energy intake alone failed to improve nutritional and functional status. However, when adequate protein intake was supplemented with essential amino acids, improvements in exercise output, peak oxygen consumption, and walking test scores were observed [6].

It has been recently demonstrated that CHF patients have lower concentrations of amino acids (AAs), particularly essential AAs (EAAs), with an inverse relationship to heart failure severity [23]. Oral supplementation with branched-chain amino acids was found to improve serum albumin levels in hospitalized patients with heart failure [24].

The important role of serum albumin in the development of heart failure was underscored in a multicenter study of community-dwelling adults without heart failure: baseline hypoalbuminemia was associated with increased risk of incident heart failure during the 10-year follow-up [25].

In the body, muscle proteins and circulating proteins (albumin) comprise the major reservoir of AAs. In the presence of malnutrition, in an attempt to maintain the energy (via gluconeogenesis, or as intermediates of the Krebs cycle) and the appropriate level of AAs (mainly the EAAs) to meet the body’s needs, the breakdown of muscle proteins occurs, thereby inducing sarcopenia [2,12,26]. Thus, in malnutrition, the reduction of caloric intake is not the main culprit responsible for tissue and organ damage. Indeed, it is interesting to note that, in mice models of qualitative malnutrition, it has been demonstrated that lifespan is inversely correlated with the percentage of non-EAAs (NEAAs) within the diet. Independent of the caloric intake, either mild limitation of EAAs intake or mild excess in NEAAs intake may induce permanent body consumption until death [27]. Pancreatic enzymes are responsible for the digestion of dietary proteins to obtain all AAs. However, the pancreas itself needs a large amount of AAs and energy to produce enzymes. Malnutrition and the hypercatabolic state may progressively reduce the efficiency of the pancreas and the mesenteric circulation, thereby impairing protein digestion and AA absorption [28]. Consequently, AA plasma concentrations decrease and may become insufficient to maintain protein synthesis and sustain energy requirements [3]. Unlike whole proteins, free AAs, provided through nutritional supplements, do not need to be digested; they are rapidly absorbed and therefore immediately available in the bloodstream for cellular uptake [3]. It has indeed been demonstrated that the availability of adequate dietary provision of all EAAs, in the correct stoichiometric proportion, is essential for the growth, development, health, and survival of animals [27,29,30] and humans [18,31].

### Nutritional Risk Assessment

Nutritional index calculations have been studied as prognosticators of outcomes in heart failure. In a study of 1673 patients hospitalized with acute heart failure (either reduced or preserved ejection fraction), the prognostic nutritional index was independently associated with cardiovascular mortality and all-cause mortality [32]. The low prognostic nutritional index was also associated with higher in-hospital all-cause mortality and post-hospitalization 6-month mortality in elderly patients hospitalized with acute decompensated HF [33].

Similarly, in elderly patients hospitalized with heart failure with preserved ejection fraction (ejection fraction ≥ 50%), a low geriatric nutritional risk index at discharge was predictive of all-cause mortality. Geriatric nutritional risk index predicts all-cause deaths in heart failure with preserved ejection fraction [34].

Malnutrition, as assessed by the Controlling Nutritional Status score on admission, was prognostic of long-term all-cause mortality in acute heart failure patients [35].

While most nutritional risk index calculators utilize serum albumin to assess the nutritional status of patients, the mini nutritional assessment screening tool may be used to identify elderly people at risk of malnutrition prior to any decline in serum albumin levels [36]. In heart failure patients in the outpatient setting, malnutrition assessed using the Mini Nutritional Assessment screening tool was predictive of mortality [37] and was an independent predictor of all-cause mortality or heart failure-related hospitalizations in outpatients with heart failure with mid-range ejection fraction [38].

## 2. Autophagy

Autophagy is a conserved cell quality control system, and increasing evidence suggests that it plays an important role in numerous and different biological processes, such as starvation, aging, inflammation, and organ remodeling, by maintaining cellular homeostasis. Autophagy occurs through the removal and degradation of intracellular components to recycle molecules needed for the synthesis of others. Three main types of autophagy have been identified: (1) macro-autophagy (commonly named autophagy), (2) micro-autophagy, and (3) chaperone-mediated autophagy. Macro-autophagy occurs by vesicle formation (autophagosome) to incorporate portions of the cytoplasm that then merge with lysosomes and result in digestion of the contents [39,40]. The presence of autophagy is demonstrated by the expression of autophagy-related proteins, such as light chain 3 (LC3), Beclin-1, autophagy-related gene (ATG), 5–12 complex, and p62 [41]. Micro-autophagy is also a degradation system that takes place directly on the surface of the lysosomes [42,43]. Finally, chaperone-mediated autophagy allows the selective degradation of only soluble proteins [40]. The prevalent function of autophagy is to promote cell survival in stressful conditions like starvation and preserve energy status in response to energy deprivation, hypoxic conditions, and high temperatures through digesting cellular components and recycling of essential elements for reuse [44,45]. Altered autophagy is also associated with many pathological states such as cancer, neurodegenerative disorders, myopathies, and cardiomyopathies [46].

## 3. Autophagy in the Heart

Autophagy in the heart occurs naturally at basal levels to maintain normal physiological cell functions [47]. However, it can be intensively activated in response to stressful situations such as decreased energy under the form of adenosine triphosphate (ATP) and increased oxidative stress, thus playing a pro-survival role [48]. During aging, although the rate of autophagosome formation and the efficiency of autophagosome-lysosome fusion as well as the proteolytic activity of lysosomes decline with age, continuous removal of exhausted or damaged components by efficient autophagy machinery, and their replacement with newly synthesized molecules, ensures cellular homeostasis and delays the aging process [48]. Indeed, it has been shown that lifelong caloric restriction by 40% increases the expression of autophagic markers in the heart [49]. The increase in autophagy resulting from caloric restriction may have a protective role on the cardiomyocytes by reducing levels of oxidative damage due to aging and cardiovascular diseases [50]. On the contrary, over-activated autophagy may deplete molecules and organelles fundamental for cellular survival, thus driving cells to death [51,52]. Autophagic degradation of “self” proteins for the production of AAs is also important for survival during neonatal starvation [53]. A state of nutrient deprivation, due to starvation or myocardial ischemia, induces autophagy, generating fatty acids and AAs that enter the Krebs cycle generating ATP and promoting survival of cardiac cells [54,55]. Although evidence indicates that physiological autophagy plays important roles in maintaining heart homeostasis, the excess of autophagy could favor and exacerbate heart-related diseases.

## 4. Autophagy in Heart Failure

Autophagy has been reported to play a role in the pathophysiology of human heart failure [46,48,56]. Many forms of heart failure are associated with the accumulation of misfolded proteins due to their impaired degradation and, hence, autophagy is fundamental under stress conditions like ischemia, starvation, and β-adrenergic stimulation, where it acts as a pro-survival mechanism by removing misfolded or anomalous proteins and damaged cellular structures [57].

Brief periods of ischemia may induce heart autophagy. In mouse hearts in vivo [54] and isolated rabbit hearts [58], autophagy was induced by ischemia and further enhanced by reperfusion. In hypoxic and re-perfused rabbit hearts, autophagosome formation has been observed after 20–40 min [59], and this is associated with functional myocyte recovery. More recently, it has been suggested that autophagy also enhances the survival of cardiomyocytes in hearts exposed to permanent coronary artery occlusion [60]. However, although many studies showed that induction of autophagy could preserve heart function during ischemia/reperfusion injury, others have suggested that autophagy contributes to cell death [41]. The magnitude of autophagy within the cell may contribute to its protective or detrimental role. Indeed, in human warm blood, cardioplegic arrest caused myocyte autophagy, with a magnitude and severity that were proportional to the length of cardioplegic arrest [61].

Autophagy has been recently extensively detected in patients with end-stage heart failure and precedes and sets the stage for the occurrence of apoptosis, oncosis, and necroptosis, which only rarely start independently of autophagy. The autophagic process progresses with broad cytosolic destruction and nuclear disintegration, finally resulting in cell death by necroptosis and, to a lesser extent, by apoptosis [62]. Then, autophagy seems to be a primary driving force leading to the progressive cardiomyocyte cell loss observed in end-stage heart failure [62]. It is interesting to note that the nucleus is the early autophagic target, and it develops typical erosions as if it had been caused by a bite (Figure 1A–C).

In advanced stages of the autophagic process, the LC3-positive vacuoles develop TUNEL positivity, indicating that DNA fragments from the nucleus have been encapsulated within autophagosomes. These data are consistent with the possibility that autophagy is a primary driving force leading to progressive cardiac cell loss in end-stage heart failure [62]. The link between CD and CHF, malnutrition, HCS, and autophagy is schematized in Figure 2.

In specific conditions, excessive activation of autophagy can lead to the destruction of essential molecules and organelles favoring cell death [50,52] (autophagy-dependent cell death) with no signs of apoptosis or necrosis. This situation was recently named autosis [63].

## 5. Role of Mammalian Target of Rapamycin (mTOR) in Autophagy

Autophagy is a dynamic process that depends on strict coordination and regulation of multiple enzymatic pathways as Beclin-1/class III phosphatidylinositol-3 kinase (PI-3K), PI-3K/Akt/mTOR pathways, and AMPK/mTOR. The last is directly influenced by the availability of nutrients, AAs in particular [64].

Usually, autophagy starts under conditions of energy deprivation in response to starvation with reduction of ATP synthesis. Adenosine monophosphate (AMP)-activated protein kinase (AMPK), an energy-sensing kinase, is activated when the concentration of ATP decreases. Under starvation (such as in ischemia or malnutrition), AMPK acts as a checkpoint by suppressing cellular growth and by promoting the activation of autophagy in cardiomyocytes. In fact, inhibition of AMPK reduces autophagy and increases cardiomyocyte cell death [55]. AMPK acts through ULK1 (Unc-51 like autophagy activating kinase) activation. In fact, ULK1 is a serine/threonine-protein kinase that mediates the induction of autophagy [65]. On the contrary, mTOR modulates autophagy by inhibiting the ULK1-kinase; it counteracts the activation of autophagy induced by AMPK. Thus, the AMPK-mTOR axis is crucial for the control of autophagy during energy stress and starvation [50]. On this basis, the excess of AMPK-induced autophagy, in a state of chronic disease or malnutrition, may be controlled by modulating mTOR activity.

mTOR is a highly conserved serine-threonine kinase that belongs to the PIKK (phosphoinositide 3-kinase related protein kinase) superfamily, which includes several kinases involved in nutrient sensing and DNA repair [66,67,68]. mTOR represents the crossroad of numerous biochemical pathways with functions that can be very different, sometimes even opposing each other. Its action, sometimes paradoxical, could be due to the fact that mTOR is the catalytic subunit of two distinct complexes: mTORC1 and mTORC2. mTORC1, localized on the outer membrane of lysosomes, and mTORC2, whose function requires association with ribosomes [69]. mTOR is activated by different stimuli, such as nutrients (nitrogen substrates provided by digestion of proteins or AAs), growth factors, energy and stress signals, and exercise. In response to these stimuli, mTORC1 mediates cell growth and proliferation. Then, mTOR acts as the center of a complex pathway network that controls protein synthesis, cell differentiation, growth, and proliferation [70,71,72]. In addition, energy variations and glucose availability are sensed by AMPK, which works coordinately with mTORC1 by shifting the cells to catabolic metabolism [73]. However, when AA availability is limited, mTOR can also operate through mTORC2 to promote autophagy [74]. Under these conditions, the role of mTORC1 shifts from suppressor to activator of autophagy, and the reactivation of mTOR is dependent on AAs one of the end products of autolysosomal degradation [75]. The different degrees of modulation of the mTORC1/mTORC2 complexes could shift a cell’s fate from survival to death and vice-versa.

It has been shown that AAs and insulin stimulate translational control of protein synthesis but in different ways. Indeed, AAs do not activate phosphoinositide-3-kinase and protein kinase B (Akt) but stimulate mTOR indirectly through TSC1/2 (tuberous sclerosis 1 and 2 alias hamartin) downregulation and Rheb (Ras homolog enriched in brain) activation in skeletal muscle of aged animals [76]. It is interesting to note that mTOR is downregulated by treatment with AAs in elderly rat hearts [64]. AAs, as well as other nutrients and physical exercise, maintain anabolism by stimulating protein synthesis in skeletal and cardiac muscles through the phosphorylation of the ribosome-associated S6 kinase. This S6-kinase activation favors a high level of translation of mRNAs that encode ribosomal proteins, thus activating both cell entry of AAs and protein synthesis [77]. In addition, AAs repress autophagy by activating the mTOR (mTORC1-mediated) metabolic pathway [29,64,78]. Indeed, AAs are essential for mTORC1 activation. When there is nutrient availability, mTOR negatively regulates autophagy via mTORC1. On the contrary, in the setting of a deficiency of AAs, mTOR is not efficiently activated by other stimuli [79,80] and can promote autophagy by operating through the mTORC2 complex [74]. The influence of AAs/nutrition, ATP/AMP, starvation, and caloric restriction on mTORC1 and autophagy is schematized in Figure 3.

## 6. The Use of EAAs in Treatment and Prevention of Malnutrition under Increased Metabolic Demand

Muscle proteins and circulating proteins, constituting the reservoir of AAs in the body, are in continuous turnover. In normal conditions, daily protein metabolism by healthy individuals consists of 250–350 g of proteins, but under conditions of increased metabolic demand (CHF, senescence, chronic diseases, tumors, etc.), this amount rises dramatically.

The current recommended dietary allowance for protein intake deemed necessary to preserve muscle health is about 0.8 g/kg/day [81,82,83]. However, people with acute or chronic diseases require a daily protein intake of at least 1.2–1.5 g/kg/day [82,84], whereas, in people with severe illnesses or overt malnutrition, the daily protein requirement increases to 2.0 g/kg/day [83]. To cope with the continuous contraction, the heart has a very high metabolic demand and is able to “burn” any substrate (fatty acids, glucose, ketone bodies, and AAs) to obtain the necessary energy. Among these molecules, only the AAs are totipotent, capable of being transformed into other high-energy molecules (glucose and lipids) to produce ATP [2]. However, compared to fatty acids and glucose, AAs contribute to a lesser degree to ATP generation in the heart. Nevertheless, AAs, particularly the EAAs, play a pivotal role in cardiac function because they are fundamental for the synthesis of proteins, many of the metabolic and signaling intermediates, and many cofactors. It has been estimated that the mammalian heart renews all cellular components in about 30 days [85], thus underlining the huge demand for the availability of EAAs for protein and cofactor synthesis.

Under chronic disease states, especially CHF, heart metabolism is chronically perturbed, resulting in an alteration of Ca^2+^ flux, generation of reactive oxygen species, energy deficiency and, finally, dysfunction of contractility. In this scenario, it seems logical to provide these patients with a suitable amount of all EAAs to stimulate anabolism and minimize autophagy of skeletal and cardiac muscles [18]. In elderly people, it has been demonstrated that muscles have reduced anabolic response to low doses of EAAs (below 10 g/day), whereas doses of about 10–15 g/day (with at least three g/day of L-leucine) induce protein synthesis comparable to that observed in younger adults [86]. As a consequence, it is strongly recommended that people with chronic diseases consume food rich in high-quality proteins with a higher amount of all EAAs or leucine-rich foods [87] or supplement their diet with an individualized stoichiometric mixture of all EAAs [3,6].

## 7. Perspectives

The balance between anabolism and catabolism of muscle and visceral proteins is crucial to maintain body integrity and adequate metabolism as well as to maintain the contractile function of myocytes and cardiomyocytes. From the literature, it is clear that malnutrition and autophagy are closely related, especially in hypercatabolic conditions, to the heart.

The AAs, mainly the EAAs, are key molecules that provide the cells with the building blocks needed to build other molecules and cellular components. In addition, AAs enter the Krebs cycle [2], promoting anabolism, favoring mitochondrial biogenesis, and reducing hyperautophagy [29]. Clinical data from CHF patients suggest that diet, supplemented with specific blends of EAAs, counteracts the severe alteration of the protein components of the body (qualitative malnutrition) and cellular energy impairment without impairing kidney function [88,89].

In light of this evidence, along with therapies recommended by good clinical practice (or clinical guideline-directed medical therapy), patients with CHF should be monitored to accurately evaluate their nutritional status and, if necessary, provided nutrients and EAAs in adequate quantities to counteract muscle hypercatabolism, malnutrition and cardiac damage resulting from hyperautophagy.

## Figures and Tables

**Figure 1 ijms-22-03332-f001:**
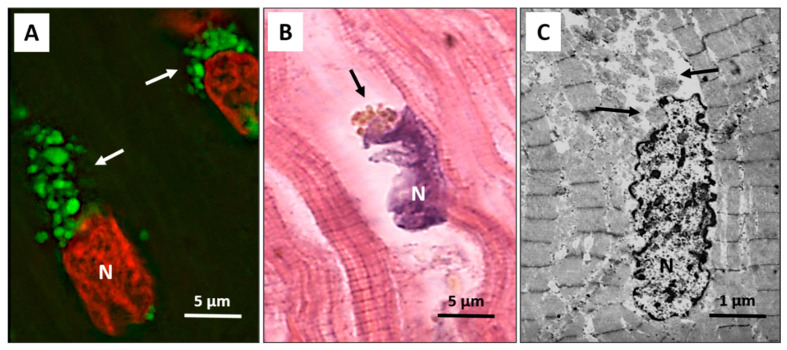
Failing heart. (**A**) Typical anti-LC3 immunofluorescence (green) of cardiomyocytes in an advanced stage of autophagic damage. Green bodies (arrows) are present around and much more at the pole of nuclei. Cardiomyocytes with massive green staining, in combination with the red staining of the nucleus, assume a “strawberry-like” appearance. N = nucleus in red; (**B**) Damaged cardiomyocytes show nuclei (N) very irregular in shape as if he had been bitten. Around these nuclei, the cytoplasm is devoid of organelles, and there are always clumps of brown debris bodies (arrow) (eosin and hematoxylin staining); (**C**) Representative transmission electron microscopy pictures show a damaged cardiomyocyte, with chromatin condensation in the nucleus (N) and deep invaginations of the nuclear envelope. Cytoplasmic granular material without organelles is present around the nucleus and near the nuclear pole (arrows).

**Figure 2 ijms-22-03332-f002:**
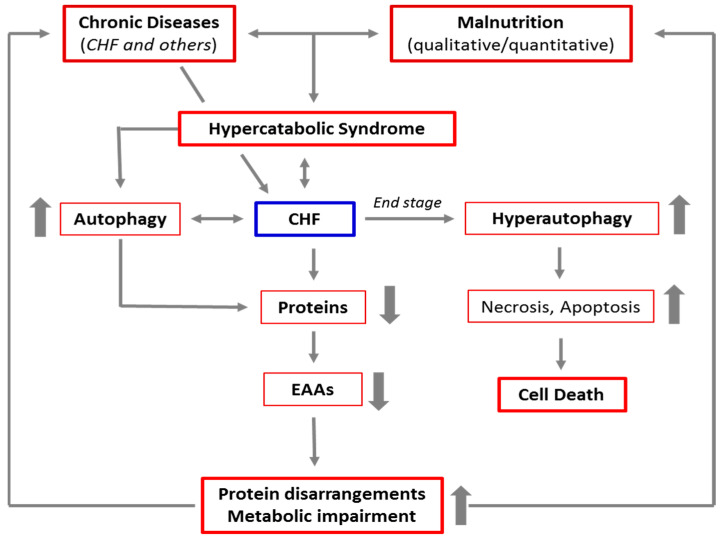
Schematic representation of the link between chronic diseases, malnutrition, hypercatabolic syndrome, and autophagy. The increase in catabolism induced by chronic diseases and malnutrition favors the activation of autophagy to guarantee the cells sufficient energy and materials to cope with accelerated metabolism. This results in a cascade of events, leading to protein disarrangements (the severe alteration of the protein components of the body), and then to metabolic impairment. This creates a vicious circle which, in the absence of adequate nutritional interventions, maintains and favors the hypercatabolic syndrome. Furthermore, in patients with advanced chronic diseases, like the end stage of CHF, autophagy is severely triggered, and the autophagic machinery may also drive the cells to self-destruction.

**Figure 3 ijms-22-03332-f003:**
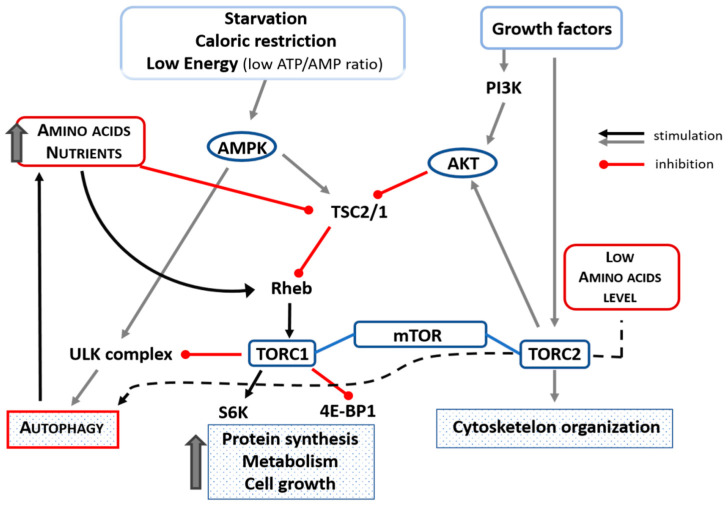
Schematic representation of the influence of amino acids, nutrients, starvation, low ATP/AMP ratio, and caloric restriction on mTORC1 and autophagy. Amino acid availability inhibits autophagy by inhibiting TSC2/1 and activating Rehb that could facilitate the transport of amino acids into the cell and which, in turn, could activate TORC1 that inhibits ULK-complex (black line). In case of low availability of amino acids, autophagy is activated through the TORC2 complex (black dotted line). Up thick arrow = increase. Akt = protein kinase B; ATP = adenosine triphosphate; 4EBP1 = eIF4E-binding protein-1; PI3K = phosphoinositide 3-kinase; Rheb = Ras homolog enriched in brain; S6 = S6-kinase; TSC1/2 = tuberous sclerosis complex 1/2; ULK (Unc-51 like autophagy activating kinase).

## Data Availability

Not applicable.

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
