# Peer review of "How Can Malnutrition Affect Autophagy in Chronic Heart Failure? Focus and Perspectives"

_ijms, 2021, doi:10.3390/ijms22073332_

Round 1
Reviewer 1 Report
This review article described malnutrition and autophagy in CHF. It is well written and useful for the readers, but I have some comments for consideration.
- I think Figures should be a key component of the manuscript and self-explanatory to some extent, i.e., readers can understand the contents without reading the text in detail. I think figure 3 is somewhat hard to understand. For example, line 254, ‘when AAs availability is limited, mTOR can also operate through the mTORC2 to promote autophagy’ is important, but connecting line of ‘TORC2’ and autophagy is absent. What is the meaning of dotted line? What are the differences between lines with ● and without? What is 4E-BP1?
- Line 51, valve dysfunction and aortic stenosis. I think aortic stenosis is one of the valve dysfunction.
- Line 54 and 214, HS should be HCS.
- Line 123, heart failure (HF) should be described when first used.
Author Response
we thank this referee for pointing out inaccuracies and for valuable suggestions.
This review article described malnutrition and autophagy in CHF. It is well written and useful for the readers, but I have some comments for consideration.
I think Figures should be a key component of the manuscript and self-explanatory to some extent, i.e., readers can understand the contents without reading the text in detail. I think figure 3 is somewhat hard to understand. For example, line 254, ‘when AAs availability is limited, mTOR can also operate through the mTORC2 to promote autophagy’ is important, but connecting line of ‘TORC2’ and autophagy is absent. What is the meaning of dotted line? What are the differences between lines with ● and without? What is 4E-BP1? We have redesigned the Figure 3 and rewrite the legend, we trust it is more intuitive, although the cascade of reactions of mTOR is very complex.
Line 51, valve dysfunction and aortic stenosis. I think aortic stenosis is one of the valve dysfunction. We agree and have changed the text deleting aortic stenosis.
Line 54 and 214, HS should be HCS. we correct it.
Line 123, heart failure (HF) should be described when first used. we correct it.
Reviewer 2 Report
Manuscript ID: ijms - 1116328
Author list Giovanni Corsetti
Evasio Pasini
Claudia Romano
Carol Chen-Scarabelli
Tiziano Scarabelli
Vincenzo Flati
Louis Saravolatz
Francesco S. Dioguardi
“How can malnutrition affect autophagy in chronic heart failure? Focus and perspectives”
SUMMARY
The manuscript is a review article on chronic heart failure (CHF) and the role that nutrition and autophagy may exert on CHF progression in patients. As in many chronic diseases, catabolism exceeds anabolism, leading to insulin resistance, muscle wasting, and an impaired ability to maintain cellular energy stores. This metabolic syndrome is associated with mortality quite apart from heart disease. Molecular targets for therapy are discussed, with a focus on pathways leading to the regulation of autophagy.
GENERAL COMMENTS
The manuscript is informative and well-referenced. The level of scientific report writing is reasonable; nevertheless, some attention to detail (both editorial and scientific) is required. To assist the authors with revisions, specific recommendations are listed below and within the manuscript. Please ignore the yellow highlight in the manuscript PDF; text underlined in green needs author attention.
TITLE
Fine
ABSTRACT
This brief text needs revision.
Protein disarrangement is an uncommon term. Please define it.
On line 22, correct the awkward wording.
“... are independent correlates of heart dysfunction, ...”
On lines 22-25, the sentence is awkward. Please restate it. An example is provided.
“Autophagy, a prominent occurrence in the heart of patients with advanced CHF, is a self-digestive process that prolongs myocardial cell lifespan by the removal of cytosolic components, such as aging organelles and proteins, and recycles the constituent elements for new protein synthesis. However, in specific conditions, excessive activation of ...”
On lines 26-27, strengthen the connection between the cellular process of autophagy and patient death. An example is provided.
“... excessive ... autophagy can lead to the destruction of molecules and organelles essential to cell survival, ultimately leading to organ failure and patient death.”
If space allows, provide relevant detail(s) about molecular markers.
Section 1
On line 54, define HS and keep abbreviations to a minimum. Similarly, on line 68, did the authors define HF? Is this abbreviation necessary? Once you define an abbreviation, use it consistently (line 118?)
On line 75, remove unnecessary punctuation.
Section 2
On lines 137-139, correct the grammar.
“Autophagy ... plays an important role in numerous and different biological processes, such as starvation, aging, inflammation, and organ remodeling, by maintaining cellular homeostasis.
On line 141, finish the thought.
“It occurs through the removal and degradation of intracellular components and subsequent recycling” ... of what?
On line 143, correct the grammar.
“... Macro-autophagy occurs by vesicle formation (autophagosomes) to incorporate portions of the cytoplasm that then merge with lysosomes and result in digestion of the contents ... “
On line 147, is “seizure” the right word?
On lines 149-151, correct the sentence structure.
“... function of autophagy is to promote cell survival ... and preserve energy status ... through digesting cellular components and recycling essential elements for reuse.”
Section 3
On lines 157-159, rephrase the sentence. Clarify how autophagy is related to aging and pro-survival. What is meant by over-regulation of autophagy?
On line 160, correct the grammar.
“... lifelong caloric restriction ...”
On lines 161-163, correct the grammar. This is a sentence fragment.
On line 165, correct the grammar.
“... On the contrary, over-activated autophagy may deplete molecules and organelles fundamental for cellular survival, thus driving cells to death.”
On lines 165-166, correct the grammar.
“Autophagic degradation of ‘self’ proteins for production of AAs is also important for survival ...”
On lines 167-171, reword these sentences using standard English.
Section 4
Paragraph 1 needs editorial revision.
In paragraph 2, on lines 187, clarification is required.
“The protective or detrimental role of autophagy may (partly?) depend on the extent of what? ... inside the cell.”
On lines 188-189, editorial revision is required.
“... in human warm blood, cardioplegic arrest caused myocyte autophagy, with a magnitude and severity that were proportional to the length of cardioplegic arrest.”
On line 197, what is meant by a bite-like appearance?
On line 211, rephrase the description of LC3 vacuoles and avoid dogmatic statements.
“... TUNEL positivity, indicating that DNA fragments from the nucleus have been encapsulated within autophagosomes. These data are consistent with the possibility that autophagy is a primary driving force leading to ...”
On lines 215-217, correct the grammar.
Section 5
On line 233, define ULK1.
On line 242, trim the text for conciseness.
“... superfamily, which includes several kinases involved in ...”
On line 262, define Akt, TSC1/2, and Rheb.
On line 263, please provide a reference.
On line 264, the plural form of AAs is not suitable. Change the sentence as follows.
“... mTOR is down regulated by treatment with AAs in elderly rat heart.”
On lines 267-268, the same applies. Correct the grammar as follows.
“... S6-kinase activation favours a high level of translation of mRNA that encode ribosomal proteins, thus activating both cell entry of AAs and protein synthesis.”
On line 274, correct the grammar.
“The influence of AAs/nutrition, ATP/AMP, starvation, and caloric restriction on mTORC1 and autophagy is ...”
Section 6
On line 286, is it necessary to define yet another abbreviation (RDA)? Keep acronyms to a minimum.
On line 293, correct the word choice.
“... capable of being transformed into other high energy molecules (glucose and lipids) ...”
On line 296, there are missing words.
“... play a pivotal role in cardiac function because they are fundamental for ...”
On line 299, correct the grammar.
“... underlining the huge demand for availability of EAAs for protein and cofactor synthesis.”
On line 302, what is metabolic inflexibility?
On line 303, do you need to define an abbreviation for reactive oxygen species? Rephrase as follows.
“... generation of reactive oxygen species, energy deficiency, and finally ...”
Section 7
On line 317, or perhaps elsewhere in the manuscript where appropriate, deliberately explain what is meant by qualitative malnutrition.
In the second paragraph, the first sentence is a run-on sentence. To add clarity, break the concepts down into at least two sentences.
On line 323, correct the grammar and clarify what “protein disarrangement means.
“... data from CHF patients suggest that diet, supplemented with specific blends of EAAs, counteracts protein disarrangement (?) and cellular energy impairment ...”
On lines 326-327, revise the sentence structure.
“... patients with CHF should be monitored to accurately evaluate their nutritional status and, if necessary, provided nutrients and ...
ACKNOWLEDGEMENTS
Fine
REFERENCES
Correct the typo in reference 36.
Remove the space between references 50 and 60.
In reference 91, does the title need correcting?
“... myocardial function in the elderly: there is a role for ...”
Reference 93 is missing from the bibliography.
FIGURES
Figure 1
In the legend, use full sentences to describe each figure panel. Cite published work and omit the photo credit. Authorship is not in question.
In panel a, what type of staining indicates the nucleus?
In panel b, “bite-like signs” is not a helpful description (in contrast to the “strawberry-like” appearance of fluorescent cells in panel a); please reword the appearance of H&E staining. Can you draw a comparison to healthy cardiomyocytes?
In panel c, fully describe the micrograph.
“Representative transmission (?) electron microscopy shows a damaged cardiomyocyte, with chromatin condensation in the nucleus and invaginations of the nuclear envelope. ...”
Figure 2
Overall, this is a useful figure. Please describe it fully in the legend. As it stands, the legend is too brief.
What are protein disarrangements? Word choice?
In the figure legend, abbreviations have already been defined. Why redefine CHF but not EAAs?
Figure 3
In the figure, “amino acids” should be two words.
Define 4E-BP1.
In the legend title, correct the grammar.
“Schematic representation of the influence of AAs, nutrients, starvation, ATP/AMP, and caloric restriction on mTORC1 and autophagy.”
Provide a full description of the figure in the legend.

Author Response
We thank this referee for the precise and accurate revision of the text and for the valuable suggestions provided. We gladly welcomed all the advice provided and the corrections suggested.
the list of corrections is attached
